# Efficient Production of the PET Radionuclide ^133^La for Theranostic Purposes in Targeted Alpha Therapy Using the ^134^Ba(p,2n)^133^La Reaction

**DOI:** 10.3390/ph15101167

**Published:** 2022-09-21

**Authors:** Santiago Andrés Brühlmann, Martin Kreller, Hans-Jürgen Pietzsch, Klaus Kopka, Constantin Mamat, Martin Walther, Falco Reissig

**Affiliations:** 1Helmholtz-Zentrum Dresden-Rossendorf, Institute of Radiopharmaceutical Cancer Research, Bautzner Landstraße 400, D-01328 Dresden, Germany; 2Technische Universität Dresden, Faculty of Chemistry and Food Chemistry, D-01062 Dresden, Germany

**Keywords:** macropa, lanthanum-133, actinium-225, PET, targeted alpha therapy, theranostics

## Abstract

Targeted Alpha Therapy is a research field of highest interest in specialized radionuclide therapy. Over the last decades, several alpha-emitting radionuclides have entered and left research topics towards their clinical translation. Especially, ^225^Ac provides all necessary physical and chemical properties for a successful clinical application, which has already been shown by [^225^Ac]Ac-PSMA-617. While PSMA-617 carries the DOTA moiety as the complexing agent, the chelator macropa as a macrocyclic alternative provides even more beneficial properties regarding labeling and complex stability in vivo. Lanthanum-133 is an excellent positron-emitting diagnostic lanthanide to radiolabel macropa-functionalized therapeutics since ^133^La forms a perfectly matched theranostic pair of radionuclides with the therapeutic radionuclide ^225^Ac, which itself can optimally be complexed by macropa as well. ^133^La was thus produced by cyclotron-based proton irradiation of an enriched ^134^Ba target. The target (30 mg of [^134^Ba]BaCO_3_) was irradiated for 60 min at 22 MeV and 10–15 µA beam current. Irradiation side products in the raw target solution were identified and quantified: ^135^La (0.4%), ^135m^Ba (0.03%), ^133m^Ba (0.01%), and ^133^Ba (0.0004%). The subsequent workup and anion-exchange-based product purification process took approx. 30 min and led to a total amount of (1.2–1.8) GBq (decay-corrected to end of bombardment) of ^133^La, formulated as [^133^La]LaCl_3_. After the complete decay of ^133^La, a remainder of ca. 4 kBq of long-lived ^133^Ba per 100 MBq of ^133^La was detected and rated as uncritical regarding personal dose and waste management. Subsequent radiolabeling was successfully performed with previously published macropa-derived PSMA inhibitors at a micromolar range (quantitative labeling at 1 µM) and evaluated by radio-TLC and radio-HPLC analyses. The scale-up to radioactivity amounts that are needed for clinical application purposes would be easy to achieve by increasing target mass, beam current, and irradiation time to produce ^133^La of high radionuclide purity (>99.5%) regarding labeling properties and side products.

## 1. Introduction

Targeted Alpha Therapy (TAT) is an emerging field in radiopharmaceutical sciences. Besides the already FDA- and EMA-approved Xofigo^®^ containing ^223^Ra in its ionic dichloride form, which is applied for late-stage and palliative treatment of bone-metastatic prostate cancer [1,2], several preclinical and early clinical trials are running on examining the application of ^225^Ac, ^212^Bi/^213^Bi and the ^212^Pb/^212^Bi in vivo generator [3], ^227^Th [4,5,6], or ^211^At [7,8,9,10] as alternative alpha emitters. Especially in the last two decades, ^225^Ac became the radionuclide of highest interest for TAT applications [11,12,13,14] because of its perfect nuclear properties (ca. 10 days of half-life, cascade decay via four alpha decays, and two beta-minus conversions). Furthermore, the trivalent actinoid [^225^Ac]Ac^3+^ ion is more easily chelated compared to the alkaline earth metal ion [^223^Ra]Ra^2+^, for example. The increasing availability of ^225^Ac in comparison to other alpha emitters is also remarkable because of its several production routes [15], which also facilitate the scaling up of the production in order to meet the increasing demand that has not been completely achieved yet. Although the generator-based approach starting with ^229^Th covers 95% of the current demand, interesting alternative routes exist, including the low-energy cyclotron-based production through the ^226^Ra(p,2n)^225^Ac reaction and the ^226^Ra(γ,n)^225^Ra → ^225^Ac route [13,16].

The nuclear properties of alpha-emitting radionuclides highlight the necessity of the highest possible complex stability in vivo when applied as pharmaceuticals. Because of the high linear energy transfer compared to beta emitters, alpha emitters are most effective when selectively bound to the biological target with high affinity. Coincidently, alpha emitters putatively have fewer side effects because of the shorter penetration depth in healthy tissue neighboring the targeted tissue. Nevertheless, the higher energy raises the issue of worse secondary effects when the formed radiometal complex is not as stable as needed in vivo. The now chelator-free radionuclide can then accumulate in the bone marrow or other organs with high cellular turnover, in general leading to unwanted accumulation in off-target regions [17].

The commonly applied chelator DOTA (1,4,7,10-tetraazacyclododecane-1,4,7,10-tetrayl)tetraacetic acid), widely used for complexation of the trivalent radiometals for clinical purposes, such as ^68^Ga, ^111^In, ^90^Y, and ^177^Lu, is also suitable for the chelation of actinium ions to a certain extent. Nonetheless, several preclinical studies indicate that there is a substantial instability at some point in vivo leading to a ^225^Ac accumulation in the liver and bones [18,19]. To overcome this obstacle, several research groups are currently working on alternative chelating agents [20]. A few years ago, the chelator *N*,*N*′-bis[(6-carboxy-2-pyridyl)methyl]-4,13-diaza-18-crown-6, also known as macropa, was introduced as a complexing agent for trivalent metal ions and showed advantages over DOTA regarding complexation behavior (lower amount of chelator needed for radiolabeling and mild radiolabeling conditions) as well as higher in vivo complex stability by not causing any unwanted radioactivity accumulation [21]. Compared with the DOTA-modified conjugates, which can be easily radiolabeled with diagnostically applicable ^111^In or ^68^Ga, a well-matching diagnostic counterpart for ^225^Ac-macropa-radioconjugates is still not clinically established.

During the last years, several approaches to the production of radioactive lanthanum isotopes have been published. Lanthanum is, from the chemical point of view, an ideal mimetic and already used for the prediction of ^225^Ac complexation behavior in a nonradioactive environment [22]. Especially, the production and imaging properties of the radionuclides ^132^La (positron emitter), ^133^La (positron emitter), and ^135^La (Meitner–Auger electron emitter) have been investigated by irradiation of a natural barium target or an enriched ^135^Ba target, leading to preliminarily satisfying amounts of the respective radionuclides in a mixture [23]. Moreover, the first phantom and in vivo experiments were carried out, leading to the assumption of radioactive lanthanum isotopes being an ideal counterpart for theranostic purposes in ^225^Ac-TAT [24,25].

However, the question may arise of how comparable are the physical properties of ^133^La with respect to other short-lived radiometals used in PET-imaging acquisition. Addressing this concern, the relevant decay properties of some positron emitters are shown in Table 1 [26]. The requirements for the acquisition of PET images with high resolution and low noise are basically a low-energy positron emitter with high intensity and low emission of high gamma energy. In recent years, scandium radionuclides, i.e., ^43^Sc and ^44^Sc, have proven to be an interesting alternative to the well-established positron emitter ^68^Ga on the basis of their lower positron energies, although the emission of higher energy gammas still needs to be addressed [27,28]. ^61^Cu also offers comparable properties to the scandium nuclides [29]. Another case study is on ^45^Ti, having suitable decay properties but also slightly more complicated coordination chemistry because of its two possible main oxidation states (+3 and +4) and potential oxidation to TiO_2_ [30]. On the other hand, the lanthanum radionuclides, although having lower positron-emission yields, show attractive decay properties. Particularly, the radionuclide ^133^La comprises a lower energy positron emission than its analog ^132^La, which is also true for most of the radionuclides listed herein, with the exception of ^45^Ti. Moreover, ^133^La also bears less intense gamma lines, thus being a more suitable match for ^225^Ac.

In this work, we report on the efficient and selective production of ^133^La by irradiating a highly enriched ^134^Ba target using the ^134^Ba(p,2n)^133^La reaction. After irradiation, the barium carbonate target was worked up, the radionuclide products and side products were identified and quantified, and test radiolabeling was performed with a recently published macropa-conjugated compound on the basis of the hitherto very well-known PSMA-617 binding vector [19,31].

## 2. Results and Discussion

### 2.1. Calculation, Target Design, and Irradiation

The irradiated targets consisted of a silver backing disc filled with [^134^Ba]BaCO_3_ with an area density of 47 mg/cm^2^, which were capped with a thin foil (10 µm platinum or 100 µm aluminum). On the basis of the ^134^Ba(p,2n)^133^La reaction cross section [32], these targets (three targets) were irradiated for an hour with 22 MeV proton beams at beam currents ranging between 10 µA and 15 µA. An aluminum degrader (0.6 mm) was used to reduce the proton energy on the target. The degraded energy in this aluminum layer, in the foil, and the [^134^Ba]BaCO_3_ was calculated by SRIM, a simulation tool to calculate the energy loss of ions in the matter [33]. The estimated energies resulted in (18.6 ± 0.1) MeV and (17.9 ± 0.1) MeV for the incident beam energy at the target material and the exciting energy, respectively.

In Figure 1, the cross sections taken from the TALYS-based evaluated nuclear data library (TENDL-2019) [32] of the relevant nuclear reactions weighted for the [^134^Ba]BaCO_3_ enriched target are displayed. Based on these cross sections, the energies previously described were chosen in order to maximize the production of ^133^La, avoiding the coproduction of other lanthanum radionuclides, i.e., ^132^La and ^135^La via the ^134^Ba(p,xn) reaction.

The coproduction of ^134^La can be neglected because of its short half-life of only 6.45 min. Furthermore, the coproduction of ^132^La is avoided by carefully choosing the incident energy of the proton beam. Additionally, small quantities of ^135^La are expected because of its long half-life, so that importance is gained after some decay time. In this case, the ^133^La-activity yield was compromised in order to ensure a ^132^La-free product.

The theoretical yield *A_EOB_* for the production of ^133^La can be calculated with Equation (1), where *N_A_* stands for the Avogadro constant, *I* for the proton beam current in µA, *M_r_* is the molecular weight of the target compound ([^134^Ba]BaCO_3_) in g/mol, *q_e_* is the charge of the electron in µC, *E_in_*, and *E_out_* the incident and exciting energy in MeV, *σ* is the cross section of the reaction in cm^2^ (taken from TENDL-2019 [32]), *S*(*E*) is the stopping power of the material in MeV cm^2^/g (taken from SRIM [33]), *t_irr_* is the irradiation time, and *T*_1/2 is_ the physical half-life of ^133^La. From this equation, and considering the previously described parameters, an activity of approx. 190 MBq/µA ^133^La was expected after one-hour irradiation.
(1)$$A_{EOB}=\frac{N_{A}\cdot I}{M_{r}\cdot q_{e}\cdot 10^{-6}}\cdot \int_{E_{in}}^{E_{out}}\frac{\sigma \left(E\right)}{S\left(E\right)}dE\cdot \left(1-2^{-\frac{t_{irr}}{T_{1/2}}}\right)$$

Activities between 1.3 GBq and 1.9 GBq of ^133^La at the end of bombardment (EOB) were reached from the target irradiation depending on the beam current. The linear correlation of the activity and the proton beam current was confirmed. The achieved ^133^La-activity yield of ca. 130 MBq/µA∙h was comparable to the theoretical yield for this target, 190 MBq/µA∙h. The difference between these yields can be attributed to several factors, such as some dispersion of the proton beam reducing the resulting current at the target, eventual uncertainties in the reported cross-section, or the loss of target material before dissolution.

The activities reached could be easily increased without compromising the product quality by modifying the target and the irradiation parameters. As displayed in Figure 1, the energy range used for this reaction was quite small and could be extended by increasing the target mass, thus increasing the activity yield of the reaction (e.g., theoretical 400 MBq/µA∙h for 60 mg of [^134^Ba]BaCO_3_ irradiated with 21 MeV protons). It is important to notice that the ^133^La/^135^La ratio should not be affected by this. Notably, increasing the beam current would also result in higher activities, as it was already seen when increasing from 10 µA to 15 µA (1.3 GBq to 1.9 GBq at EOB). Last but not least, one-hour irradiation is still far from saturation, which offers the possibility of carrying out longer irradiation times which would lead to higher yields.

### 2.2. ^133^La Product Characterization

After irradiation, the solid target was directly transferred for separation and manually opened. The powder was separated from the target disk and foil and dissolved in 3 mL of 1 M HNO_3_. An initial sample was collected for γ-spectroscopy, the calculation of the ^133^La yield, and the amounts of coproduced side products. The gamma spectrum and the characteristic gamma energy lines of the raw product solution are examples shown in Figure 2.

Since both radionuclide and isotope impurities cannot be initially detected (because of the overall small number of impurities), a second gamma spectrum of the same solution was recorded 24 h after EOB to quantify leftovers. This gamma spectrum and the marked impurity peaks are displayed in Figure 3.

Quantification and identification of the product and side products were carried out by high-purity germanium (HPGe) gamma spectrum analyses. The following relative amounts of radionuclides were detected in the raw solution (measured 25 min after EOB and calculated for EOB): ^133^La (99.5%), ^135^La (0.4%), ^135m^Ba (0.03%), ^133m^Ba (0.01%), and ^133^Ba (0.0004%). The value of ^135m^Ba was calculated during the separation process, as follows.

### 2.3. ^133^La Purification and Characterization

The applied purification process was conducted according to a published procedure by Wuest et al. [25] with slight adjustments. The previously dissolved target material was directly loaded on a preconditioned (3 M HNO_3_) branched cartridge with diglycolamide (DGA) resin. Afterward, 50 mL of 3 M HNO_3_ were automatically eluted through the column, and a sample was collected every 3 to 5 mL for exact quantification of eluted products. The purification scheme is displayed in Figure 4.

The shown purification process leads to a reliable ^133^La separation and also enables the possibility of target material recycling. To better understand the separation process, samples were analyzed by HPGe radiation detection, and the following elution profile was determined (Figure 5).

A straightforward and very sharp separation of lanthanum and barium isotopes was achieved using this method. According to the displayed elution profile, the first 10 mL of process solution were collected for the ^134^Ba recovery process, which could be conducted in the second step, e.g., by carbonate precipitation in high yields. The product fraction was collected in 5 × 1 mL aliquots, and the highest amount of ^133^La was found in the second milliliter (rel. amount > 85%, concentration ≥ 1 GBq/mL, necessary for radiolabeling in smaller volumes). Within this separation process—taking ca. 30 min in total—it was possible to collect both target material fractions and product fractions greater than 95% decay-corrected to EOB in small volumes, as shown in Figure 5, which can be used for either radiolabeling without further processing or target recovery. However, further optimization of the separation will be carried out in order to reduce the elution volumes and the produced volume for target recovery. A remaining amount of 0.04 kBq of ^133^Ba per 1 MBq of ^133^La was detected 72 h after the radiochemical separation caused by the ^133^La/^133^Ba decay scheme (Figure 6). This small amount was valued as not relevant in any context of waste management or radiation protection concerns.

### 2.4. Radiolabeling with [^133^La]La^3+^

As a proof of concept, the radiolabeling with ^133^La was performed using the previously published compound **mcp-M-PSMA** [31], a macropa-derived conjugate based on the PSMA-617 binding vector, which has already been investigated with respect to the evaluation of the pharmacological behavior of the appropriate ^225^Ac-radioconjugate **[^225^Ac]Ac-mcp-M-PSMA** expressed as biodistribution in mice. For this purpose, 5 MBq of ^133^La were radiolabeled quantitatively with **mcp-M-PSMA** when applying ligand concentrations of ≥1 µM in 0.2 M ammonium acetate solution (pH 6) for 15 min at room temperature. These values correspond to our already well-established in-house radiolabeling of **mcp-M-PSMA** with [^225^Ac]Ac^3+^. Radiolabeling reactions were monitored by a radio thin-layer chromatograph in a 50 mM EDTA solution of neutral pH value. The complete radiochemical conversion was obtained when applying ligand concentrations ≥ 5 µM. A radio-HPLC chromatogram to characterize the radiolabeled product **[^133^La]La-mcp-M-PSMA** and to determine the radionuclide purity is displayed in Figure 7.

The chromatogram indicates a complete complexation of [^133^La]La^3+^ combined with a high radiochemical purity of the formed radiolabeled complex **[^133^La]La-mcp-M-PSMA** (ca. 98%). One side product was formed as well, but it is neglectable because of the small relative amount of ≤2% and was not further characterized.

The maximum of achievable molar activity was determined to be ca. 330 GBq/µmol (regarding the used amount of ligand **mcp-M-PSMA**). The labeling results are consistent compared to our already performed studies with the ^225^Ac-labeled conjugate [31]. Compared with other diagnostically used tracers, the value is rated as high, especially for radiometal-based conjugates. For comparison, an apparent molar activity ≥ 18.5 GBq/µmol to 35.5 GBq/µmol is usually exhibited for clinically applied [^68^Ga]Ga-PSMA-11 [34,35].

## 3. Materials and Methods

### 3.1. Target Preparation

Silver discs (2 mm thickness with 22 mm diameter) with a deepening (0.3 mm depth with 9 mm diameter) were filled with 30 mg [^134^Ba]BaCO_3_ and capped with a 10 µm platin foil. [^134^Ba]BaCO_3_ of the isotopic composition shown in Table 2 was supplied by Isoflex. After loading, the target was pressed with a hydraulic press. In the following irradiation, the platin foil was replaced with a 100 µm aluminum foil in order to reduce activation products on the foils as well as the operating costs. The disc and foil materials were chosen in order to ensure the optimal cooling of the target while not producing huge amounts of activation products.

Regarding the target material, enriched [^134^Ba]BaCO_3_ was chosen on the bases of its favorable cross section of the desired nuclear reaction ^134^Ba(p,2n)^133^La. In Figure 8, the cross section of different nuclear reactions leading to La isotopes from a natural barium target and the enriched ^134^Ba target are illustrated [29]. Results coming from Figure 8 motivate the use of enriched material in order to increase the radionuclide purity of the product. Target recycling of the enriched [^134^Ba]BaCO_3_ is envisaged.

### 3.2. Cyclotron Irradiation

The target irradiation was carried out at the TR-FLEX (ACSI) cyclotron at the HZDR, using the 90° configuration and the afore-described solid target. One-hour irradiation starting with a 10 µA beam current was performed in order to ensure the target safety, increasing the current to 15 µA for the next irradiations.

### 3.3. Radiochemical Separation of ^133^La

The target was opened manually, the powder was dry-separated by gravity and subsequently dissolved in 3 mL of 1 M HNO_3_. The solution was loaded on a preconditioned (10 mL 3 M HNO_3_) branched DGA cartridge (Triskem, cartridge volume 2 mL, 760 mg). Then, the DGA cartridge was washed with 50 mL of 3 M HNO_3_ (automatically eluted via syringe pump, Harward Apparatus; flow rate approx. 3 mL/min) and deacidified with 5 mL of 0.5 M HNO_3_. In the last step, the [^133^La]La^3+^ product was eluted with 5 × 1 mL of 0.05 M HCl. The first 10 mL of the total 50 mL washing solution were collected for later ^134^Ba recovery.

### 3.4. Radionuclide Characterization

Radionuclide identification and quantification were carried out by high-resolution gamma spectroscopy using an energy- and efficiency-calibrated Mirion Technologies (Canberra) CryoPulse 5 HPGe detector. Each sample was diluted to a total amount of 200 µL and poured into a plastic tube with calibrated geometry for the gamma spectroscopy measurement. The radionuclides were identified by comparing gamma lines with the radionuclide database, and activity values were calculated using the respective efficiency calibration function, both automatically via the software Genie2000 (V. 3.4.1).

### 3.5. ^133^La-Radiolabeling and Quality Control

Radiolabeling was performed using **mcp-M-PSMA** as a test ligand. A total of 5 MBq of ^133^La (in 0.05 M HCl) were poured into an Eppendorf DNA low-bind tube, 100 pmol of ligand (10 µL of a 10^−5^ M stem solution in 0.2 M ammonium acetate, pH 6) were added, and the total reaction volume was filled up to 100 µL with 0.2 M ammonium acetate solution (pH 6). The reaction mixture was shaken at 500 rpm in an Eppendorf Thermomixer at room temperature for 15 min. Afterward, 1 µL sample was taken for subsequent radio-TLC analyses on normal phase Silica plates (Merck). The TLC plate was placed on a radiation-sensitive film, and the film was imaged by the laser-mediated Phosphorimager (Amersham Typhoon), whereas in free La^3+^ run with the front in EDTA solution (50 mM, pH 7), the radiolabeled complex **[^133^La]La-mcp-M-PSMA** remained completely at the origin. 

The formed complex **[^133^La]La-mcp-M-PSMA** (approx. 100 kBq of the radiolabeling mixture) was further characterized by radio-HPLC to determine the product purity using a Jasco HPLC system connected to the Gabi Detector for radioactivity measurement, a Phenomenex Kinetex Biphenyl (100 mm × 4.6 mm) column as stationary phase and water/acetonitrile as mobile phase (gradient 5–95% acetonitrile in 9 min, 0.1% TFA each, flow rate: 1 mL/min.). Labeling and quality control were performed in accordance with a previously published procedure for labeling with ^225^Ac [31].

## 4. Conclusions

In conclusion, an efficient production and purification route for ^133^La using a ^134^Ba target was presented, which complements the already existing methods via ^nat^Ba and ^135^Ba irradiation, but delivers an advantageous product composition regarding ^133^La radiochemical yield and purity. Only 0.4% of coproduced ^135^La was detectable, which seems neglectable for future (pre-)clinical applications. The amount of ^133^La using the presented production parameters is high enough for numerous preclinical studies or up to one or two patient doses but can be improved by varying the irradiation parameters. Coproduced and decay product amounts of long-lived ^133m/g^Ba are rather uncritical both in the target and the product leftovers. ^133^La yield could be easily increased by adjusting the beam current, time, and target mass which is used in order to supply a large amount of this PET radionuclide (approx. up to 10 patient doses) to hospitals for clinical investigations. Thereby, ^133^La opens the gate for a perfect matching theranostic application of future ^225^Ac-therapeutics carrying the macropa chelator instead of DOTA.

## Figures and Tables

**Figure 1 pharmaceuticals-15-01167-f001:**
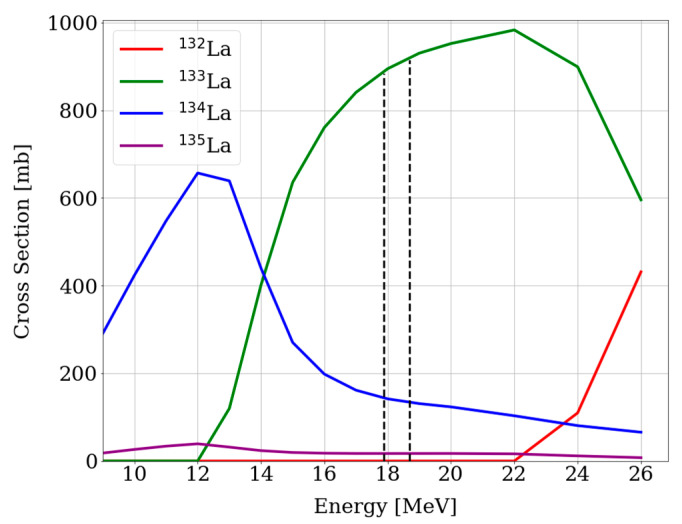
Calculated cross sections of relevant nuclear reactions using TENDL-2019, leading to La radioisotopes via the ^134^Ba(p,xn) route weighted for the [^134^Ba]BaCO_3_ enriched target. The dotted lines indicate the energy range used in this work.

**Figure 2 pharmaceuticals-15-01167-f002:**
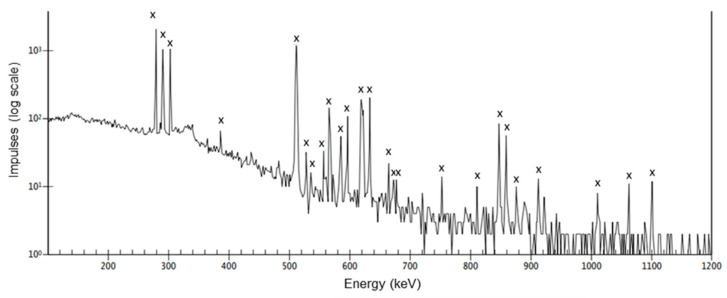
Representative gamma spectrum of the raw target solution 25 min after EOB. X—characteristic gamma lines for ^133^La.

**Figure 3 pharmaceuticals-15-01167-f003:**
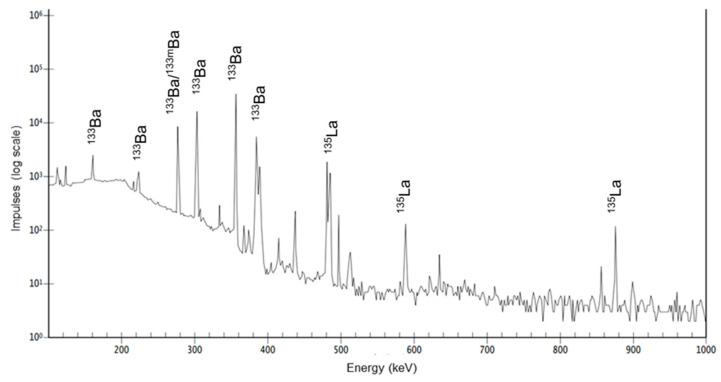
Representative gamma spectrum of the raw target solution 24 h after EOB and marked characteristic gamma lines of ^133m/g^Ba and ^135^La impurities.

**Figure 4 pharmaceuticals-15-01167-f004:**
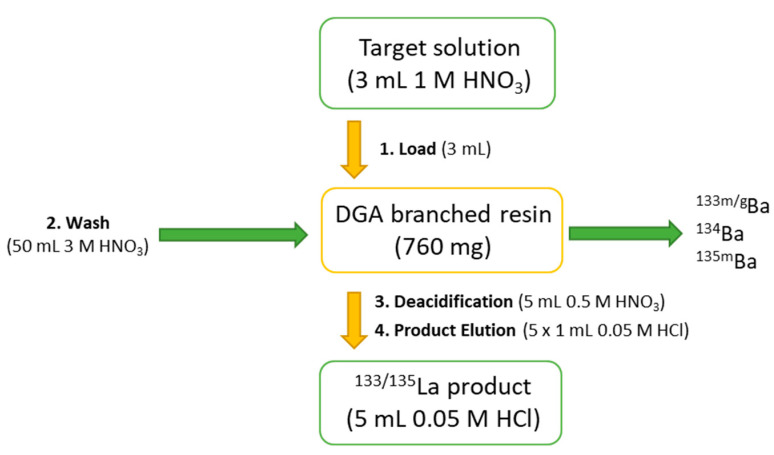
^133^La purification scheme.

**Figure 5 pharmaceuticals-15-01167-f005:**
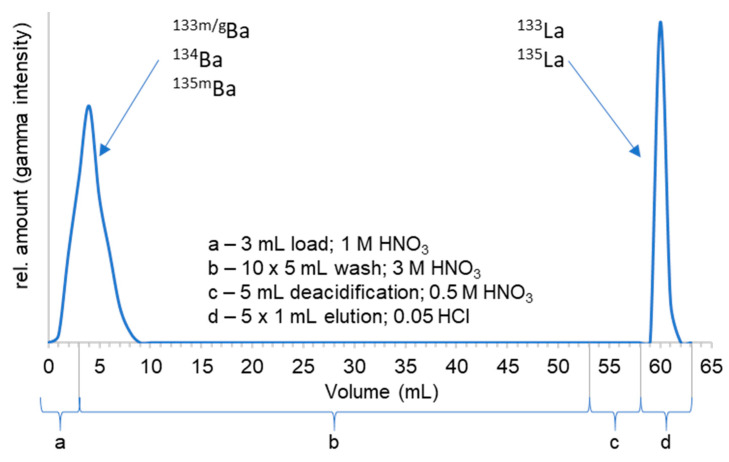
Elution profile of the Ba/La separation process using the DGA cartridge.

**Figure 6 pharmaceuticals-15-01167-f006:**
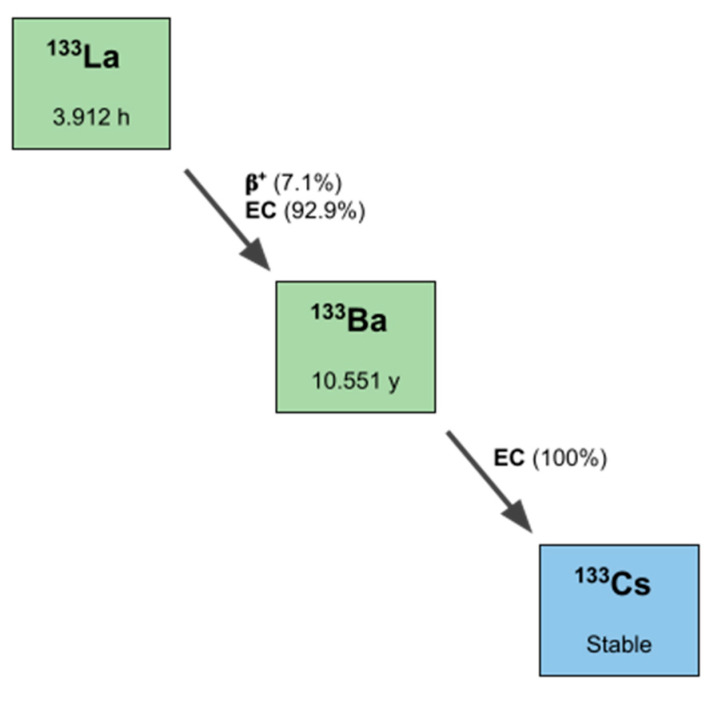
Decay scheme of ^133^La, including the half-life and type of decay.

**Figure 7 pharmaceuticals-15-01167-f007:**
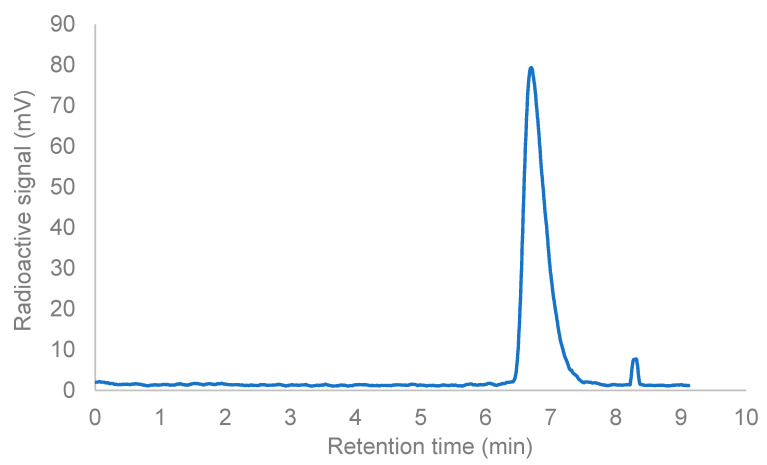
HPLC chromatogram of **[^133^La]La-mcp-M-PSMA** (radioactive signal vs. retention time).

**Figure 8 pharmaceuticals-15-01167-f008:**
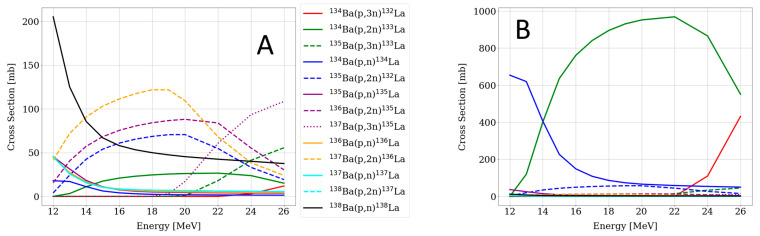
Cross section of nuclear reactions leading to La radioisotopes weighted for (**A**) natural BaCO_3_ and (**B**) enriched [^134^Ba]BaCO_3_ target material.

**Table 1 pharmaceuticals-15-01167-t001:** Physical properties of short-lived β^+^-emitting radiometals suitable for PET imaging.

Radionuclide	Half-Life	E_β+,mean_/keV (Intensity/%)	E_γ_/keV (Intensity/%)
^43^Sc	3.89 h	508 (70.9)	372.9 (22.5)
		344.5 (17.2)	
^44^Sc	3.97	632 (94.3)	1157 (99.9)
^45^Ti	3.08	439 (84.8)	no γ-line >1%
^61^Cu	3.34	524 (51)	282.9 (12.7)
		399 (5.8)	656.0 (10.4)
		238 (2.5)	67.41 (4.0)
		494 (2.1)	1185 (3.6) i.a.
^68^Ga	1.13	836 (87.72)	1077 (3.22)
		352.6 (1.19)	
^132^La	4.8	1454 (14)	464.5 (76)
		1191 (11)	567.1 (15.7)
		1665 (9.2)	1909 (9.0)
		496 (2)	663.0 (9.0)
		582 (1.4) i.a.	1031 (7.8) i.a.
^133^La	3.91	463 (7.1)	278.8 (2.44)
			302.4 (1.61)
			290.1 (1.38)
			12.3 (1.38)

**Table 2 pharmaceuticals-15-01167-t002:** Isotopic composition of the irradiated [^134^Ba]BaCO_3_ as specified by the supplier.

Isotope	^130^Ba	^132^Ba	^134^Ba	^135^Ba	^136^Ba	^137^Ba	^138^Ba
Content [%]	<0.01	<0.01	88.10 ± 0.40	5.36	1.21	1.07	4.26

## Data Availability

Not applicable.

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
