# Peer review of "Efficient Production of the PET Radionuclide 133La for Theranostic Purposes in Targeted Alpha Therapy Using the 134Ba(p,2n)133La Reaction"

_pharmaceuticals, 2022, doi:10.3390/ph15101167_

Round 1

Reviewer 1 Report

The authors describe in this manuscript the production of lanthanum-133 via a cyclotron-based method and the processing/purification steps of the resulting radionuclides. For exemplification, 133La obtained this way was engaged in the radiolabeling of a macropa-containing PSMA derivative. The authors can be congratulated for the quality and significance of their work, which is well exposed and focuses on a major topic of modern nuclear medicine. Indeed, the increasing trend towards the clinical use of alpha emitters, particularly from the actinide and lanthanide groups, is driving the search for surrogates atoms and new theranostic pairs. The article is clear and structured, focusing on a topic that fits well to the scope of the journal. Suitable bibliographical references illustrate this work properly.

I have a few remarks and suggestions to bring to the attention of the authors:

- Line 16: please write “DOTA core” or “DOTA moiety”.

- Line 16, 19, 29 and so on: the word “macropa” is sometimes written in italics, sometimes not. Please be consistent in the writing, as there is no obvious reason for this word to appear in italics.

- Line 26: please do not use abbreviations in the abstract and write “end of bombardment” in full-letters.

- Line 38: please write the ® symbol in superscripts.

- Line 40 to 52: please group the bibliographic references as it won’t appear, for example, as “[4][5]” but as “[4,5]”.

- Line 47: Actinium-225 is indeed becoming more and more widely produced in the world, but the term "high availability" may seem exaggerated for this radioelement (ref 13 states: "The current supply of most of the TAT radionuclides is insufficient for pre-clinical and clinical evaluation"). Please either document further this "high availability" with recent bibliographic references or consider slight rewording.

- Line 55 to 58: this sentence is quite long, and the affinity/selectivity part is unclear. Please consider rewording.

- Line 116: maybe a photography of this target would be interesting. This comment is left to the discretion of the authors.

- Line 133: Although 134La displays a short half-life, what about its daughters (radio)nuclides? Don't they induce cold metal impurities in the produced solution? Can these impurities be extracted? Please further discuss this point.

- Line 148: please write “from the target” without capital letter.

- Line 164: maybe writing “which would lead to higher yields” would be more suitable.

- Line 192: maybe writing “diglycolamide (DGA) resin” and not only “DGA resin” would be informative; this comment is left to the discretion of the authors.

- Paragraph 1.4: beyond in-house experience, are the radiolabeling results obtained here consistent with those already published with macropa-containing vectors? Please discuss this point at the end of the radiolabeling description, with appropriate bibliographic references.

- Line 239: since the analysis is mentioned above, please add a short sentence about the values obtained with the TLC method. Is it also possible, for information only, to give the radiochemical yield values calculated with both methods?

- Table 2: for the value of 134Ba, please add a space before and after “±”.

- Lines 277-278: if the reader does not refer to Figure 5 simultaneously, this sentence is quite unclear, as the elution as such appears to result in 5 mL HCl 0.05 M. Therefore, the origin of the “first 10 mL for 134Ba recovery” is not easy to understand. Please consider slight rewording.

- Paragraph 3.5: is there any reference publication for these radiolabeling conditions? If yes, please state.

- Line 288 and 289: please specify in which solvent initially are the 5 MBq of 133La and the 100 pmol of ligand, respectively.

- Conclusion: at the beginning of the conclusion part, the “already existing methods” for production and purification of 133La are mentioned, however, these techniques of production and processing/purification should be further discussed all along the manuscript and compared to the results of this work.

- Line 312: just like at line 164, it would probably be more suitable to write “could be easily increased” as it has not been formally verified.

Author Response

Answers to Report 1

The authors describe in this manuscript the production of lanthanum-133 via a cyclotron-based method and the processing/purification steps of the resulting radionuclides. For exemplification, 133La obtained this way was engaged in the radiolabeling of a macropa-containing PSMA derivative. The authors can be congratulated for the quality and significance of their work, which is well exposed and focuses on a major topic of modern nuclear medicine. Indeed, the increasing trend towards the clinical use of alpha emitters, particularly from the actinide and lanthanide groups, is driving the search for surrogates atoms and new theranostic pairs. The article is clear and structured, focusing on a topic that fits well to the scope of the journal. Suitable bibliographical references illustrate this work properly.

We thank the reviewer a lot for his/her fruitful and warm comments and honouring our work. We have carefully revised our manuscript step by step including the reviewer’s comments to a major extent.

I have a few remarks and suggestions to bring to the attention of the authors:

- Line 16: please write “DOTA core” or “DOTA moiety”.

Moiety was added.

- Line 16, 19, 29 and so on: the word “macropa” is sometimes written in italics, sometimes not. Please be consistent in the writing, as there is no obvious reason for this word to appear in italics.

It is not appearing any more in italics.

- Line 26: please do not use abbreviations in the abstract and write “end of bombardment” in full-letters.

This is corrected.

- Line 38: please write the ® symbol in superscripts.

Has been changed.

- Line 40 to 52: please group the bibliographic references as it won’t appear, for example, as “[4][5]” but as “[4,5]”.

We have checked and corrected the references.

- Line 47: Actinium-225 is indeed becoming more and more widely produced in the world, but the term "high availability" may seem exaggerated for this radioelement (ref 13 states: "The current supply of most of the TAT radionuclides is insufficient for pre-clinical and clinical evaluation"). Please either document further this "high availability" with recent bibliographic references or consider slight rewording.

Rewording performed – term changed to “The increasing availability…that has not been completely achieved, yet.”

- Line 55 to 58: this sentence is quite long, and the affinity/selectivity part is unclear. Please consider rewording.

Intense rewording was performed.

- Line 116: maybe a photography of this target would be interesting. This comment is left to the discretion of the authors.

We would prefer not to share photographs of the target.

- Line 133: Although 134La displays a short half-life, what about its daughters (radio)nuclides? Don't they induce cold metal impurities in the produced solution? Can these impurities be extracted? Please further discuss this point.

Lanthanum-134 decays to Barium-134 (100%) – Barium-134 is separated during the work-up, since it is identical with the target material and separated anyway. A separate paragraph was not included.

- Line 148: please write “from the target” without capital letter.

Corrected.

- Line 164: maybe writing “which would lead to higher yields” would be more suitable.

We have changed the expression as suggested.

- Line 192: maybe writing “diglycolamide (DGA) resin” and not only “DGA resin” would be informative; this comment is left to the discretion of the authors.

We have inserted the explanation.

- Paragraph 1.4: beyond in-house experience, are the radiolabeling results obtained here consistent with those already published with macropa-containing vectors? Please discuss this point at the end of the radiolabeling description, with appropriate bibliographic references.

An appropriate statement was added.

- Line 239: since the analysis is mentioned above, please add a short sentence about the values obtained with the TLC method. Is it also possible, for information only, to give the radiochemical yield values calculated with both methods?

A statement was added to the manuscript.

- Table 2: for the value of 134Ba, please add a space before and after “±”.

Corrected.

- Lines 277-278: if the reader does not refer to Figure 5 simultaneously, this sentence is quite unclear, as the elution as such appears to result in 5 mL HCl 0.05 M. Therefore, the origin of the “first 10 mL for 134Ba recovery” is not easy to understand. Please consider slight rewording.

We tried to clarify it by rewording and naming it the first 10 mL of total 50 mL washing solution.

- Paragraph 3.5: is there any reference publication for these radiolabeling conditions? If yes, please state.

Reference was inserted.

- Line 288 and 289: please specify in which solvent initially are the 5 MBq of 133La and the 100 pmol of ligand, respectively.

Both solvents were added to the experimental section.

- Conclusion: at the beginning of the conclusion part, the “already existing methods” for production and purification of 133La are mentioned, however, these techniques of production and processing/purification should be further discussed all along the manuscript and compared to the results of this work.

We have tried to refer to existing work, whenever possible. The main differences are focussed on product purity and composition, not the separation process itself. The different targets that are used as well as a similar production procedure are, at least in our opinion, confidently cited.

- Line 312: just like at line 164, it would probably be more suitable to write “could be easily increased” as it has not been formally verified.

Corrected.

Report 2

The present paper is an interesting research focused on the production of 133La, its purification and labeling with PSMA-inhibitors by the chelator macropa.

The research is focused on a very cutting-edge topic, that is the applications of targeted alpha therapy in prostate cancer, and provides a relatively easy and reproducible approach to obtain 133La-conjugated compounds.

I congratulate with the authors for the interesting paper. The research protocol is meticulously carried out.

I kindly suggest to cite the following papers in the introduction, when the authors deal with 225Ac-PSMA and Thorium compounds:

    DOI: 10.1089/cbr.2019.3105

    DOI: 10.1080/14737140.2020.1814151

Thank you very much for your kind comments. We considered your suggestions and included these references in the respective introduction sections. They are now numbered as references [6] and [14].

Reviewer 2 Report

The present paper is an interesting research focused on the production of 133La, its purification and labeling with PSMA-inhibitors by the chelator macropa.

The research is focused on a very cutting-edge topic, that is the applications of targeted alpha therapy in prostate cancer, and provides a relatively easy and reproducible approach to obtain 133La-conjugated compounds.

I congratulate with the authors for the interesting paper. The research protocol is meticulously carried out.

I kindly suggest to cite the following papers in the introduction, when the authors deal with 225Ac-PSMA and Thorium compounds:

  • DOI: 10.1089/cbr.2019.3105
  • DOI: 10.1080/14737140.2020.1814151

Author Response

Answers to Report 2

The present paper is an interesting research focused on the production of 133La, its purification and labeling with PSMA-inhibitors by the chelator macropa.

The research is focused on a very cutting-edge topic, that is the applications of targeted alpha therapy in prostate cancer, and provides a relatively easy and reproducible approach to obtain 133La-conjugated compounds.

I congratulate with the authors for the interesting paper. The research protocol is meticulously carried out.

I kindly suggest to cite the following papers in the introduction, when the authors deal with 225Ac-PSMA and Thorium compounds:

    DOI: 10.1089/cbr.2019.3105

    DOI: 10.1080/14737140.2020.1814151

Thank you very much for your kind comments. We considered your suggestions and included these references in the respective introduction sections. They are now numbered as references [6] and [14].